# Chlorination of *Schistosoma mansoni* cercariae

**Laura Braun** [1]*, **Yasinta Daniel Sylivester** [2], **Meseret Dessalegne Zerefa** [3], **Muluwork Maru** [3], **Fiona Allan** [4], **Feleke Zewge** [3], **Aidan M. Emery** [4], **Safari Kinung'hi** [2], **Michael R. Templeton** [1]

**1** Department of Civil and Environmental Engineering, South Kensington Campus, Imperial College London, London SW7 2AZ, United Kingdom, **2** National Institute for Medical Research, Mwanza Center, Isamilo Street, Ilemela, Mwanza, Tanzania, **3** College of Natural and Computational Sciences, Addis Ababa University, Arat Kilo, Addis Ababa, Ethiopia, **4** Wolfson Wellcome Biomedical Laboratories, Departsssment of Life Sciences, Natural History Museum, Cromwell Road, London SW7 5BD, United Kingdom

* laura.braun13@imperial.ac.uk

## Abstract

### Background

Schistosomiasis is a water-based disease acquired through contact with cercaria-infested water. Communities living in endemic regions often rely on parasite-contaminated freshwater bodies for their daily water contact activities, resulting in recurring schistosomiasis infection. In such instances, water treatment can provide safe water on a household or community scale. However, to-date there are no water treatment guidelines that provide information on how to treat water containing schistosome cercariae. Here, we rigorously test the effectiveness of chlorine against *Schistosoma mansoni* cercariae.

### Method

*S. mansoni* cercariae were chlorinated using sodium hypochlorite under lab and field condition. The water pH was controlled at 6.5, 7.0 or 7.5, the water temperature at 20˚C or 27˚C, and the chlorine dose at 1, 2 or 3 mg/l. Experiments were conducted up to contact times of 45 minutes. 100 cercariae were used per experiment, thereby achieving up to 2-$\log_{10}$ inactivations of cercariae. Experiments were replicated under field conditions at Lake Victoria, Tanzania.

### Conclusion

A CT (residual chlorine concentration x chlorine contact time) value of 26±4 mg·min/l is required to achieve a 2-$\log_{10}$ inactivation of *S. mansoni* cercariae under the most conservative condition tested (pH 7.5, 20˚C). Field and lab-cultivated cercariae show similar chlorine sensitivities. A CT value of 30 mg·min/l is therefore recommended to disinfect cercaria-infested water, though safety factors may be required, depending on water quality and operating conditions. This CT value can be achieved with a chlorine residual of 1 mg/l after a contact time of 30 minutes, for example. This recommendation can be used to provide safe water for household and recreational water activities in communities that lack safe alternative water sources.

**Data Availability Statement:** All relevant data are within the manuscript.

**Funding:** This research is funded by Merck KGaA (LB), the Engineering and Physical Sciences Research Council (EPSRC) Global Challenges

Research Fund (GCRF) via grant reference EP/
P028519 (LB, YS, MD, MM, FA, AME, SK, FZ,
MRT), and the EPSRC Centre for Doctoral Training
in Sustainable Civil Engineering via grant reference
EP/L016826/1 (LB). The funders had no role in
study design, data collection and analysis, decision
to publish, or preparation of the manuscript.

**Competing interests:** Merck KGaA provide partial
sponsorship of Laura Braun's PhD bursary and
tuition fees. Merck KGaA had no role in study
design, data collection and analysis, decision to
publish, or preparation of the manuscript. There
are no competing interests or other links with
Merck KGaA.

## Author summary

Schistosomiasis is a preventable yet serious disease in many developing countries.
Although schistosomiasis control strategies include the provision of water and sanitation
facilities, specific WASH solutions are often lacking. Control programs generally focus on
preventive chemotherapy with praziquantel which is effective in killing adult worms in
humans but does not prevent re-infection. Communities lacking safe alternative water
sources therefore have no choice but to rely on unsafe water for their daily water contact
activities, even if this exposes them to high risk of infection. Water treatment can provide
communities with safe water on a household or community scale. However, to-date there
are no guidelines for treating cercaria-contaminated water. We rigorously tested the effec-
tiveness of chlorine against *S. mansoni* cercariae in lab and field settings. The resulting
chlorination recommendation can be used to treat water and provide schistosomiasis-
endemic communities with safe water facilities (e.g. laundry or bathing stations).

## Introduction

Schistosomiasis is a water-based neglected tropical disease. It affects approximately 200 million
people worldwide and targets the most marginalized communities that are reliant on contami-
nated water sources [1]. The disease is contracted through contact with cercaria-infested fresh-
water. These cercariae burrow through the hosts' skin to develop into schistosome worms,
leading to schistosomiasis infection. Aquatic snails, living in freshwater bodies such as rivers
or lakes, release cercariae. Snails are infected by miracidia hatched from eggs released through
urine or faeces of infected people. Effective deworming medication is available; however, this
does not prevent re-infection which is inevitable if daily water activities are reliant on contact
with contaminated water (e.g. bathing, laundry) [2].

It is evident that the provision of clean water plays a pivotal role in schistosomiasis control,
and that treatment alone will not break the transmission cycle [3]. Several studies have shown
a direct link between water contact and infection rates [4–6], and a recent study found that for
every percentage increase in piped water coverage, there was a 4.4% decline in *Schistosoma
haematobium* re-infection intensity [7]. However, no water treatment guidelines for schistoso-
miasis-endemic regions exist to-date, and our recent review found that there was insufficient
data to develop recommendations [8]. This makes it difficult to implement safe water infra-
structure in endemic areas and has hindered the progress of Water Sanitation and Hygiene
(WASH) for preventing schistosomiasis infection. Hence, there is a need to develop water
treatment recommendations by rigorously testing the effectiveness of water treatment pro-
cesses to remove or inactivate schistosome cercariae. Here, we test the effectiveness of chlori-
nation against *Schistosoma mansoni* cercariae, one of the three main species that infect
humans. This widespread *Schistosoma* species is found in Africa, Asia and the Americas, and
leads to intestinal schistosomiasis [2].

Chlorination is an effective water treatment process used anywhere from small-scale emer-
gency settings (e.g. chlorination tablets) up to large-scale municipal water treatment plants.
Since its first use in public water supplies in the 1900s, it has contributed to significant reduc-
tions in water-borne diseases [9]. Chlorine can be dosed with a chemical dosing pump, tablets
or manual dosing and stirring, for example. Chlorine kills pathogens through oxidation which
disintegrates the cell wall. It thereby alters the permeability of the cellular envelope, interferes
with membrane functions, impairs enzyme and protein functions, and denatures nucleic acid

[10]. The predominant disinfection mechanism is pathogen-specific and could be a combination of all these processes. When chlorine is added to water, it reacts with organic material (vegetation, animals, ammonium, decaying organic matter and organic nitrogen) and forms chloramines. The amount of chlorine consumed by the water is termed the chlorine demand; e.g. the more organic material, the higher the chlorine demand of the water. The remaining unreacted chlorine is called the residual free chlorine and is available for disinfection. It forms hypochlorite ions and hypochlorous acid in water. The ratio of these species is determined by the pH of the water. Hypochlorous acid, a strong disinfectant, is preferentially formed under acidic conditions, and hypochlorite ions under alkaline conditions. As a result, chlorination is more effective under acidic conditions [10]. Chlorination is also influenced by water temperature which slightly speeds up the chemical reaction resulting in more effective chlorination at higher temperature.

A 'CT' value is used as a comparative measure to indicate how sensitive organisms are to chlorine. It is the product of the residual chlorine concentration (C) and contact time (T). If a pathogen requires higher CT values to be inactivated to the same log inactivation as another pathogen, this indicates higher chlorine tolerance. CT values are also used as a water treatment process monitoring parameter, meaning that pathogen monitoring is not necessary as long as the required CT value is maintained. The WHO considers a pathogen's resistance to chlorine to be low for CT values <1 mg·min/l, moderate for 1–30 mg·min/l, and high for >30 mg·min/l (for 2-$\log_{10}$ inactivation at 20˚C and pH 7–8) [11]. Their guidelines for drinking water treatment recommend free chlorine residuals ≥0.5 mg/l after at least 30 minutes, i.e. CT values ≥15 mg·min/l to treat drinking water of low turbidity (<10 NTU) [11]. The effectiveness of chlorination (i.e. the CT value) is directly related to water pH and temperature, and these variables must therefore be controlled in experiments.

We systematically reviewed the effect of chlorine on schistosome cercariae [8] and found that CT value recommendations in the literature varied between 3–30 mg·min/l, depending on cercarial concentration, pH, temperature, water matrix, species and chlorine form. The lack of reporting of many of these key variables, especially the chlorine residual concentration, was a major weakness of many previous studies. In the present study we have therefore run a series of controlled chlorination experiments, evaluating the effect of pH (pH 6.5–7.5) and temperature (20–27˚C) on the effectiveness of chlorination at killing *S. mansoni* cercariae.

## Materials and methods

### Ethics statement

The complete life cycle of *S. mansoni* NMRI (Puerto Rican) strain is maintained at the Wellcome Sanger Institute. All the regulated procedures were conducted under the Home Office Project Licence No. P77E8A062, and protocols were revised and approved by the Animal Welfare and Ethical Review Body (AWERB) of the WSI. The AWERB is constituted as required by the UK Animals (Scientific Procedures) Act 1986 Amendment Regulations 2012.

### Snail infection

*S. mansoni* infected mouse livers were obtained from the Wellcome Sanger Institute (UK). Livers were collected from mice infected with 250 cercariae and perfused 40 days post-infection. Schistosome eggs were extracted using a Pitchford funnel [12] and hatched into miracidia in bottled water. Uninfected *Biomphalaria glabrata* snails were obtained from the Schistosomiasis Collection at the Natural History Museum (SCAN), London, UK. One snail was placed in a beaker with 20 ml water and 10 miracidia for 12 hours to ensure each snail was equally infected (as opposed to batch-infecting snails). To provide enough cercariae and allow for snail

mortality, at least 25 snails were infected each month. Snails were kept at 27˚C at the Natural History Museum (London, UK), and cercarial production started three to four weeks post-infection.

## Water characteristics in lab experiments

Bottled drinking water was used in all experiments. Instead of buffering bottled water to adjust the pH, three types of bottled water were selected based on their pH: pH 6.5, 7.0 and 7.5 (Buxton, Volvic and Highland Springs, respectively). The pH of every bottle was measured before experiments and varied by pH±0.1 of the value indicated on the label. This water was used for snail handling (rinsing, shedding) as well as the respective chlorination experiment. Water temperature was controlled with a water bath at 20˚C and 27˚C which is within the temperature range of optimal schistosome transmission by *Biomphalaria glabrata* snails [13]. Although air temperature in schistosomiasis endemic regions fluctuates below the tested range of 20–27˚C, water temperatures rarely drop below this. The water temperature of Lake Victoria ranges between 23.5–29.0˚C, for example [14, 15]. Furthermore, *Biomphalaria glabrata* snails have been shown to die if kept at temperatures below 16˚C [16]. Therefore, water temperatures below 20˚C were not tested. To confirm that the water did not influence the cercarial lifespan, cercariae were left for 24 hours in the three bottled waters at 20˚C and 27˚C, after which all cercariae were still fully active.

## Cercarial preparation and enumeration

Prior to chlorination experiments, 10 snails were placed in the dark for at least 38 hours to induce cercarial shedding. The snails were then rinsed, placed in a beaker with 20 ml bottled water, and exposed to visible light (11W LED lamp). After 60 minutes, snails were removed, and the cercaria-infested water was filtered through a 200 μm polyester mesh to remove snail faeces. To estimate the concentration of cercariae, three 100 μl aliquots were taken by pipette and 10 μl of Lugol iodine solution (Sigma Aldrich) was added. This instantly stained and killed cercariae, which could then be easily counted under a stereo microscope. The average cercarial concentration per aliquot was calculated, and the cercarial solution was diluted accordingly with water to achieve the desired concentration (100 cercariae/3 ml) and used immediately for experiments run at 27˚C. For 20˚C experiments, the cercariae were placed in a 20˚C water bath for 10 minutes. All experiments used fresh cercariae (<1.5 hours old), except one set of experiments in which aged cercariae were used. These were produced by storing the fresh cercarial solution in a glass-covered beaker at 27˚C for 6 hours.

Field experiments at the National Institute for Medical Research (NIMR) Mwanza Centre in Tanzania used water and *Biomphalaria sudanica* snails collected from a transmission site at Kigongo Ferry, Lake Victoria, Tanzania (approx. 2˚42'47.6"S 32˚53'37.0"E). Snails were placed in 12-well multi-well plates (Corning) filled with 5 ml bottled water and exposed to light for 30 minutes. The plates were examined under a stereo microscope and infected snails shedding only *S. mansoni* cercariae were used for experiments. Unfiltered lake water was used for these field experiments, and the temperature and pH were not controlled. The average temperature and pH were 24±0.7˚C and 7.4±0.2, respectively. The lake water was breakpoint chlorinated before the experiment, meaning the chlorine concentration exceeded the oxidant demand of the water. This ensured that a free chlorine residual was present at the start of the experiment.

## Chlorination

Every week, a fresh 50 mg/l chlorine solution was prepared by diluting reagent-grade sodium hypochlorite solution (10–15% available chlorine, Sigma-Aldrich) in distilled water. The pH,

temperature and chlorine dose were controlled in all experiments. pH was controlled by selecting bottled water with given pH (6.5, 7.0, 7.5) instead of using pH buffers which may affect cercarial lifespan. The water temperature was controlled with a water bath, and the chlorine dose added to the water (1, 2, 3 mg/l) was controlled by adding the required volume of 50 mg/l chlorine solution.

First, glassware soaked in a chlorine bath (10 mg/l) for 24 hours was rinsed three times with distilled water. Then, bottled water (43.2, 44.5, 45.6 ml, depending on chlorine dose) was added to a 100 ml beaker. Chlorine was dosed into the beaker and mixed using a glass rod. 3 ml of cercarial solution (100 cercariae/3 ml) were added to the beaker, bringing the total volume to 50 ml. The water was gently mixed using the rod and the time was started. The beaker was then placed in a 20˚C water bath for experiments run at 20˚C or left at the regulated snail culture room temperature for 27˚C experiments. After the respective contact time (5–45 minutes, in 5-minute increments), the viability of the cercarial sample was assessed by counting number of live and dead cercariae under a stereo microscope–namely moving and non-moving cercariae. The log-inactivation is calculated using Eq 1, and from here on $log_{10}$ is referred to as log. Given a sample of 100 cercariae, a 1- and 2-log inactivation are equivalent to 90% and 99% inactivation, respectively.

$$Log\ inactivation = log_{10} \frac{total\ number\ of\ cercariae}{alive\ cercariae} \qquad \text{Eq 1}$$

The chlorine residual was measured using the DPD-FAS (N,N-diethyl-p-phenylene diamine - ferrous ammonium sulfate) titrimetric method (Standard Method #4500–CL.F [17]) once the desired contact time had elapsed. The volumes of reagents and sample were scaled by 50% as the sample volume was 50 ml instead of 100 ml as used in the standard method (i.e. 2.5 ml of buffer and DPD instead of 5 ml). All chemicals used to make the reagents were purchased from Sigma-Aldrich.

### Quality control

The ferrous ammonium sulphate (FAS) solution was standardized against potassium chromate solution once a week, in duplicate [17]. If one of the test results was not within 10% of the expected value, the FAS solution was re-made and re-standardized. For quality control of the cercarial viability assessment, ten random dead cercariae were touched using a needle and assessed for movement [18]. Experiments run under the same conditions (i.e. pH, temperature, chlorine dose, contact time) were replicated at least three times.

### Results

Our results reflect low turbidity (<1 NTU) and chlorine-demand free water conditions. The CT and log inactivation data is plotted in Fig 1, showing a directly proportional relationship. In total, 18 conditions were tested (three chlorine doses, three pH values, two temperatures). For simplicity, only data of experiments run at pH 6.5 and 20˚C are shown, as all conditions followed a similar trend. The data shows shouldering and tailing, and therefore the linear Chick-Watson disinfection model does not apply [19]. Instead, the data was fitted with third degree polynomial curves, achieving $R^2$ values between 0.85–0.95 for all conditions. The tailing at high CT values may be due to variations in the population resistance to chlorine [20].

The chlorine dose was controlled at 1, 2 or 3 mg/l to determine if the resulting CT values and inactivations would differ. Fig 1 indicates that slightly lower CT values are required at lower chlorine doses. For example, for the same inactivation of 1.5 log, CT values of 13, 14 or 17 mg·min/l are required at chlorine doses of 1, 2 or 3 mg/l, respectively. This suggests that the

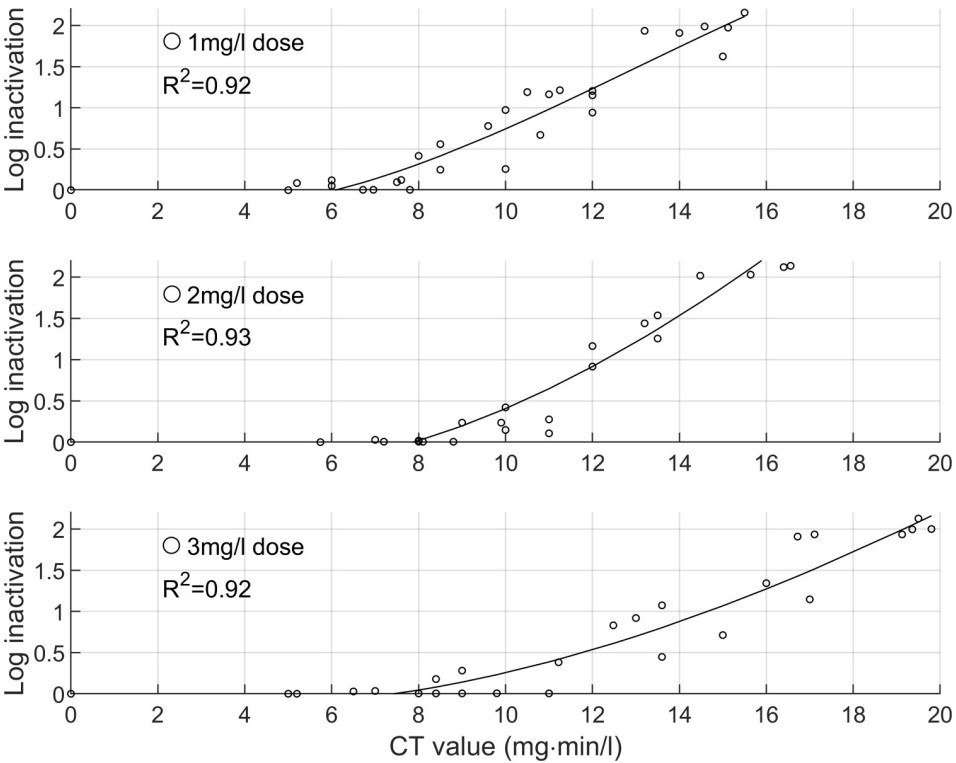

**Fig 1. Log inactivation of S. mansoni cercariae versus CT values, for chlorine dose of 1, 2, 3 mg/l.** Data shown is for pH 6.5 and 20˚C.

inactivation of cercariae by chlorine is more influenced by the chlorine contact time than the residual chlorine concentration.

To confirm this hypothesis, the time and chlorine residual required to achieve a 2-log inactivation (±0.1) were plotted (Fig 2). Each subplot contains results from all three chlorine doses. As the slope is always less than 1, it confirms that chlorine contact time is more important than chlorine residual.

The fitted polynomial curves for the different pH and temperature conditions are shown in Fig 3. Fig 3A shows inactivation at 20˚C and 3B at 27˚C. These results are for a 3 mg/l dose as this is the highest dose tested and hence produces the most conservative results (i.e. lower chlorine doses of 1 and 2 mg/l require lower CT values). Comparing Fig 3A and 3B, it is evident that slightly higher CT values are required at lower temperature for the same log inactivation and pH. This is because chemical reactions such as oxidation are faster at higher temperature, and therefore chlorination is occurring at a slightly faster rate at 27˚C than 20˚C. On average, the required CT value increases by 0.34 mg·min/l per 1˚C decrease. pH on the other hand has a much larger influence on chlorination, increasing the required CT values by an average 5.54 mg·min/l per 1.0 pH value increase. As previously explained, under alkaline conditions, sodium hypochlorite reacts to form hypochlorite ions which are a weaker chlorine species than hypochlorous acid, formed under acidic conditions. Therefore, more chlorine is needed at higher pH.

CT values required to achieve 0.5, 1.0, 1.5, and 2-log inactivations were interpolated from Fig 3, and are shown in Table 1, rounded to the nearest whole numbers. Again, the results are for a chlorine dose of 3 mg/l, since this condition produced the most conservative (highest) required CT values. The chlorine residual concentrations at 3 mg/l dose ranged between 1.3–

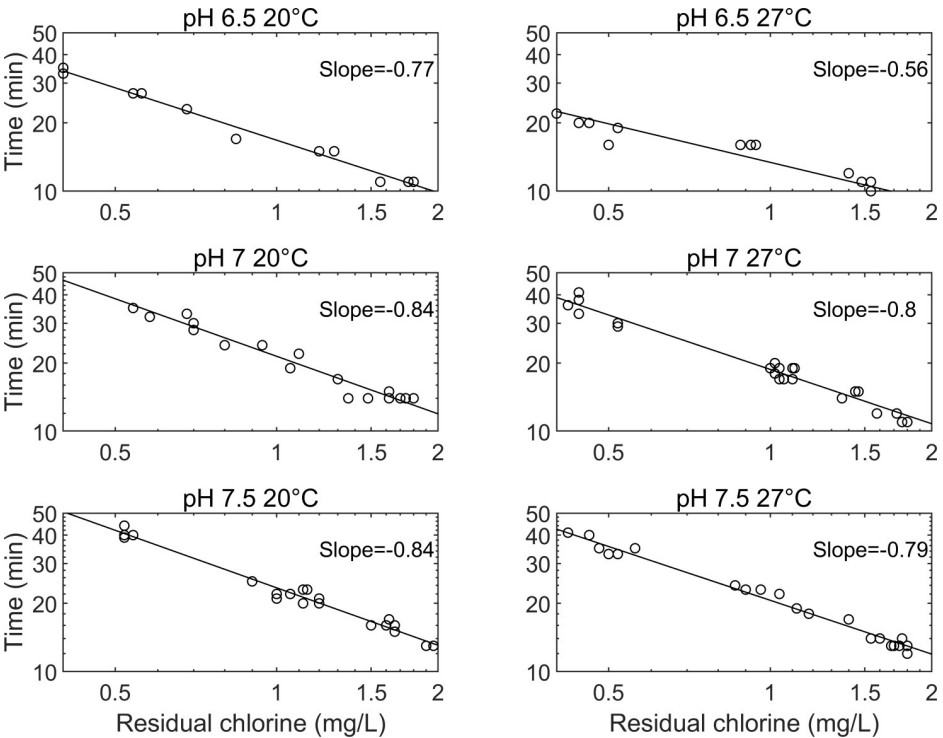

**Fig 2. Chlorine residual and time required to achieve a 2.0 ±0.1 log inactivation of S. mansoni cercariae.** Results include data for all tested chlorine doses (1,2,3 mg/l).

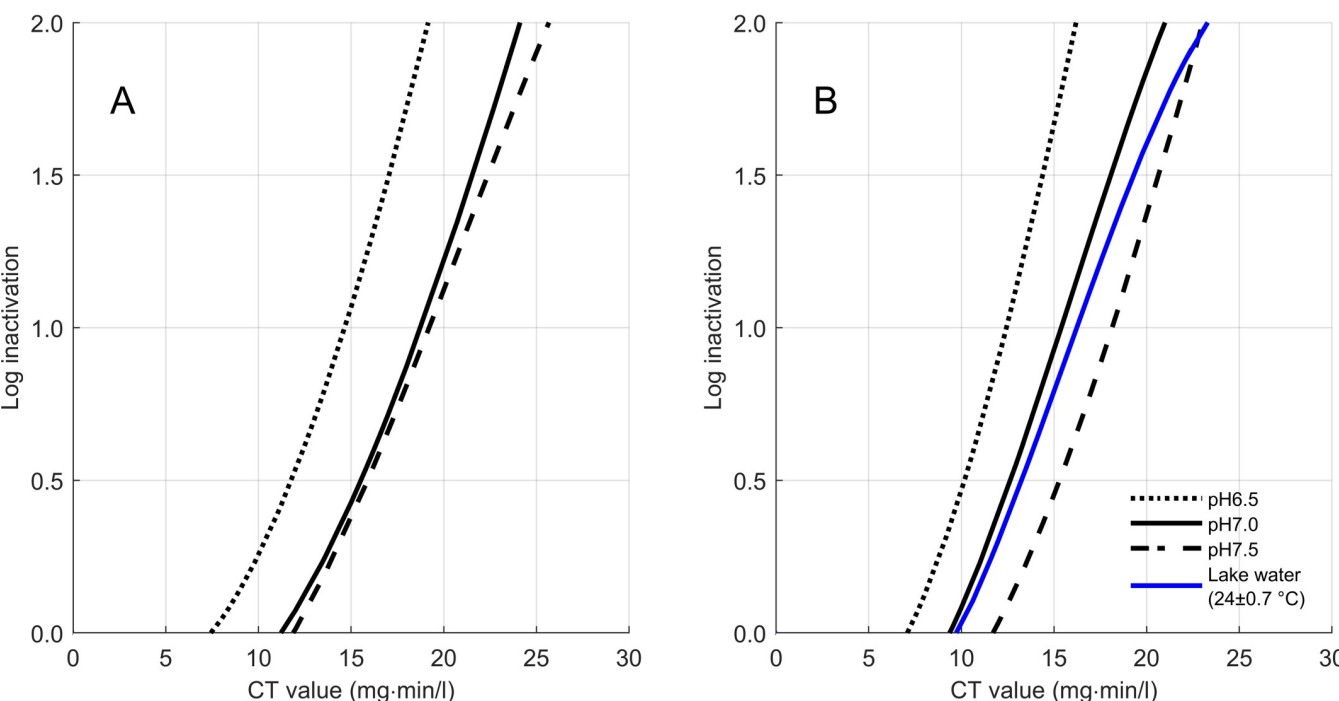

**Fig 3.** CT values required to inactivate S. mansoni cercariae at 20˚C (Fig 3A) and 27˚C (Fig 3B) in bottled water samples with pH 6.5, 7.0 or 7.5. Fig 3B also shows CT values under field conditions using water and snails collected from Lake Victoria.

**Table 1. CT-values (mg·min/l) required to achieve log$_{10}$ inactivation, based on 3 mg/l chlorine dose, which corresponded to a free chlorine residual in the range of 1.3–1.9 mg/l.**

| | 20˚C | | | 27˚C | | |
|---|---|---|---|---|---|---|
| Log$_{10}$ Inactivation | pH 6.5 | pH 7.0 | pH 7.5 | pH 6.5 | pH 7.0 | pH 7.5 |
| 0.5 | 12 | 16 | 16 | 11 | 13 | 16 |
| 1.0 | 15 | 19 | 20 | 13 | 16 | 19 |
| 1.5 | 18 | 22 | 23 | 15 | 19 | 21 |
| 2.0 | 20 | 25 | 26 | 17 | 21 | 24 |

1.9 mg/l in the bottled water experiments, reflecting the chlorine demand exerted by the cercariae themselves. This is within the WHO recommended range for drinking water and therefore does not require dechlorination [11, 21]. The chlorine residual concentrations are also below the organoleptic thresholds for chlorine in water (<2mg/l for odor) [21]. As expected, the lowest CT values were achieved at pH 6.5 and 27˚C, and the highest CTs at pH 7.5 and 20˚C. Overall, the highest CT value (and hence the most conservative CT recommendation) was 26 mg·min/l for a 2-log inactivation of *S. mansoni* cercariae. The uncertainty in measurements at this log-inactivation is estimated at 2.5 mg-min/l, arising from errors in measurements and the accuracy of equipment (e.g. burette, pH meter). The standard deviation is 1.5 mg-min/l, due to random experimental variability. Therefore, the overall uncertainty in the CT value to achieve 2-log inactivation of cercariae is estimated at 4 mg-min/l.

The CT value can be expressed as a regression equation in terms of the free chlorine residual at time $T$ (*Residual*), pH, temperature, and empirical constants *a*, *b*, *c*, *d* (adapted from Clark *et al.* [22]). Solving the multivariate regression using our data for 2-log inactivations at varying water pH and temperature, the equation becomes:

$$CT = 0.8271 \times Residual^{0.2203} \times pH^{2.3605} \times Temperature^{-0.4643} \qquad \text{Eq 2}$$

The pH constant has the highest value which confirms the variable's significant effect on the CT value. Temperature on the other hand has a much lower, negative, constant, representing the inversely proportional relationship with CT value. To illustrate this fit, Fig 4 shows raw data (chlorine residual and time) for 2.0±0.1 log inactivation at pH 7.5 and temp 20˚C against predicted values using Eq 2. Error bars derive from experimental variability in measuring time and residual chlorine (since the chlorine demand varied slightly between experiments) and variability in resistance to chlorine. Overall, the results indicate that the model is a good fit with $R^2$ of 0.80. The equation can be used to interpolate CT values for 2-log inactivation of cercariae within the temperature range of 20–27˚C and pH 6.5–7.5.

The results presented above were derived from experiments with laboratory-bred snails and cercariae in bottled waters. To test if the results are valid for field conditions i.e. naturally occurring schistosome infected snails and natural water, experiments were replicated at the National Institute for Medical Research (NIMR) Mwanza centre, Tanzania. The temperature and pH of the lake water was on average 24±0.7˚C and 7.4±0.2, respectively. As these exact water conditions were not tested in our laboratory experiments, it is difficult to directly compare the results. Nonetheless, Fig 3B shows that the CT values required under natural conditions are in line with those required under laboratory conditions. The results can be fit with a 3$^{rd}$ degree polynomial with $R^2$ of 0.89, as the disinfection kinetics show shouldering and tailing at low and high inactivations. This indicates that cercarial inactivation under field conditions aligns with the laboratory findings. The CT value required for 2-log inactivation under field conditions is 24 mg·min/l. To directly compare the field data and lab data at 2-log, the regression equation can be used to determine the expected CT value under lab conditions. With an

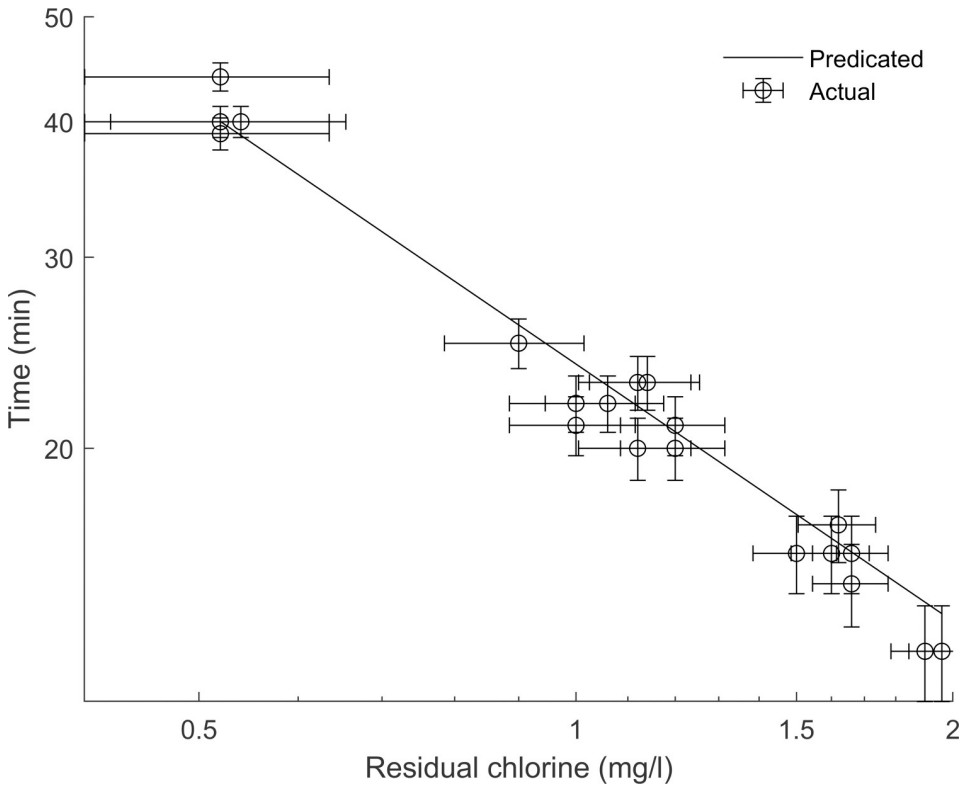

**Fig 4. Residual chlorine (mg/l) and time (min) required to achieve a 2.0±0.1 log inactivation of S. mansoni cercariae.** Data shown is for pH 7.5 and 20˚C. Eq 2 was used to yield predicted time values given actual residual chlorine values. Error bars show uncertainty in measurements arising from experimental error.

average chlorine residual of 0.87 mg/l, pH 7.4 and temperature 24˚C, the rounded CT value calculated with the regression equation is 21 mg·min/l. The CT value under field conditions is slightly higher than the predicted value under laboratory conditions. However, it cannot be concluded that field cercariae are more chlorine resistant because the field data falls within the uncertainty of the lab data, calculated as ±4 mg-min/l.

To determine if fresh and aged cercariae are equally as sensitive to chlorine, we replicated a set of experiments using aged cercariae (6 hours old) under the worst-case condition (pH 7.5, 20˚C). The average CT value required to achieve a 2-log inactivation using old cercariae is 15.7 ±0.7 mg·min/l, whereas for fresh cercariae 24.9±0.5 mg·min/l. A t-test confirms that old cercariae require statistically significant lower CT values than fresh cercariae ($p < 0.001$).

## Discussion

The results indicate that natural and laboratory-cultivated *S. mansoni* cercariae have similar chlorine sensitivities; CT values required under field conditions (water and snails collected from Lake Victoria) and those produced with the regression equation for the same pH, temperature and chlorine residual are in agreement. This result is crucial as it further suggests that the laboratory results are also valid under field conditions where this knowledge would be applied in practice. Conditions experienced by cercariae in the natural environment are different from those produced in ideal laboratory conditions. Growth condition can change a microorganism's chlorine resistance by altering the cell envelope [10], which is especially the case for bacteria. The growth condition is known to significantly affect a snail's cercarial

production [23, 24], but it is unknown if it also affects the morphology and physiology of cercariae. Cercariae are non-feeding and may therefore be less affected by the water composition compared to other microorganisms that depend on the water matrix as a growth medium. The results also indicate that physiochemical agents found under field conditions do not significantly affect the effectiveness of chlorine against cercariae, as long as the chlorine demand is properly taken into account. Such agents include clay, detritus or other particulate matter, and can provide physical protection from chlorine. This is often seen with bacteria and viruses [10]. Disinfectant protection is enhanced with decreasing organism size, since the organisms are better able to be shielded within particles. Given the large size of cercariae compared to bacteria or viruses, physical protection is less likely to affect cercarial disinfection, however this should be kept in mind when treating water with high turbidity (i.e. containing visible suspended particulate matter).

Although cercariae may be too large to be protected by most particulate matter, cercariae can protect each other from chlorine by clumping together. Rarely, cercariae were found agglutinated in large masses at the bottom of the beaker. Cercariae can release adhesive material from their acetabular glands, most likely a mucoid secretion from the post-acetabular penetration gland [25]. This may be released by cercariae under stress, caused by chlorine disinfection for example. Clumping also occurs if cercariae are too concentrated (>2000 cercariae/ml) [26]. Clumped cercariae did require higher chlorine residuals or longer chlorine contact times. This may be due to the clumping reducing the surface area of cercariae exposed to chlorine, thereby shielding cercariae at the centre of the agglomeration. This occurrence was very rare, only affecting two out of more than 500 experiments, and is potentially less likely in natural water where the cercarial concentration is much lower [27, 28].

Our recent review identified cercarial age to potentially play an important role in chlorination as cercariae became more sensitive to chlorine with age (>1hr) [8]. This is confirmed by our results which demonstrate that aged cercariae are statistically significantly less resistant to chlorine than fresh cercariae. This could be due to cercariae being released within mucus from the snail [29], which may act as a protective layer. Cercariae are non-feeding and their glycogen content declines exponentially with ageing [30], providing a further explanation as to why older cercariae are more sensitive to chlorine.

The viability of cercariae was assessed by counting number of live (moving) and dead (non-moving) cercariae post-treatment. This is the most common method for assessing cercarial viability but can be subjective [31]. To add a level of certainty, cercariae were touched with a needle to assess for movement [18]. Other water treatment experiments have used infectivity methods (e.g. suspending a mouse's tail in treated cercarial water and observing infection [32, 33]). These very accurately predict the point at which water is safe (i.e. no longer leading to infection), but due to the requirement for animal testing this method was not considered. Other methods such as the recently developed fluorescence assays [34] are not applicable as they require incubation periods of more than 10 minutes which would significantly affect the chlorine contact time. Quenching the chlorine at the start of the assay using sodium thiosulphate was considered, however the chemical visibly changed the cercariae, making them swollen and slow-moving.

Results presented here are for *S. mansoni* cercariae, which are among the most widespread human schistosome species. *S. haematobium* and *S. japonicum*, as well as species that infect livestock and have the potential for hybridization, should be tested in future research to determine inter-species variability. The cellular envelope and enzymes are often the primary target in chlorine disinfection [10]. All schistosome cercariae are covered by a glycocalyx that controls cell surface properties such as cell permeability. This, in addition to the similar morphology of schistosome cercariae, leads us to hypothesize that the disinfection mechanism and

hence the required CT values will not differ significantly for different schistosome species. Nonetheless, three studies that simultaneously tested the effect of chlorine on *S. haematobium* and *S. mansoni* cercariae found that the two species had differing sensitivities to chlorine [35–37]. All three studies were published in the early 1900's, and lacked accurate chlorine measuring equipment or did not control water conditions. Furthermore, *S. japonicum* and *S. intercalatum* tend to aggregate in clumps due to sticky secretions [38] which could potentially increase the required CT value. This highlights the need to conduct further research on all species under controlled conditions.

The results presented here can be used as the basis for water treatment guidelines for *S. mansoni* infested water. Our raw results show that a CT value of 26 mg·min/l can achieve a 2-log inactivation under our most conservative condition (pH 7.5, 20˚C). Published guidelines for CT values generally add a safety factor to the raw data and account for a degree of uncertainty. The estimated uncertainty in the data of 4 mg·min/l arises from the standard deviation as well as the variability in counting cercariae, time measurements, burette reading, pH and temperature, thereby increasing the recommended CT value to 30 mg·min/l. Typically, safety factors are additionally applied to raw data to account for variability in water composition, chlorine strength, inter- and intra-species chlorine resistance, and user errors. This can be expressed as a percentage of the raw data, or CTs for a higher log inactivation can be proposed for a lower inactivation (e.g. data for 99.9% inactivation used for 99% inactivation) [39]. However, there are several reasons why this may not be necessary for our data. Firstly, movement was used as a measure for cercarial viability, but from a public health point of view, cercariae only need to be treated to a non-infective stage. Hence, our CT values to produce 'safe' water may be overestimated. Furthermore, the recommendation of 30 mg·min/l is for a 2-log inactivation, meaning 1 cercaria survives out of a sample of 100. One cercaria is sufficient to develop into a schistosome, and two schistosomes are required to produce eggs. Therefore, water should be completely free of cercariae. This cannot be guaranteed with a 2-log inactivation which is equivalent to a 99% inactivation. However, water samples are unlikely to contain as high concentrations of cercariae as tested in our experiments (2 cercariae/ml). Therefore, a 2-log inactivation may be sufficient to inactivate all cercariae. This is unlike wastewater treatment where helminth eggs are found in much higher concentrations and therefore require higher log inactivations [40]. For example, water with a concentration of less than 1 cercaria/ 100L (as found in the field on St Lucia [41]) would enable the safe treatment of up to 10,000L, as this volume would contain less than 100 cercariae. Cercarial concentration can vary significantly in and between water sources based on the time of day, distance to shedding snails and many other factors [27, 28, 42]. It is therefore recommended that water intended for treatment is abstracted in the morning [42] away from vegetation and snail habitats to minimize the initial cercarial concentration (e.g. at the end of a jetty in a lake). Furthermore, the CT of 26±4 mg·min/l is based on a sample with only fresh cercariae, which is unlikely to occur in the field. Experiments using 6-hour old cercariae suggest that chlorine sensitivity increases with age, and hence the recommended CT value is based on conservative assumptions. Ultimately, water quality regulators and practitioners should decide whether additional safety factors should be applied to the recommendations based on the local context (e.g. water quality, cercarial concentration, water abstraction point).

The residual chlorine can be measured using a titration, or simpler tools such as pool testing kits or digital colorimeters [43]. In low-resource settings where schistosomiasis is endemic, such testing kits may not be available. We therefore recommend that chlorination guidelines should also be expressed as contact time and chlorine dose, instead of chlorine residual. One option for producing such guidelines is measuring the CT values in low turbidity water, and doubling the required chlorine dose to account for water quality variability in the field [44].

Alternatively, the chlorine residual can be converted to a chlorine dose with knowledge of the raw water's chlorine demand and the strength and form of chlorine present. Disinfection recommendations could be given for a range of chlorine demands and chlorine strengths. For example, a CT value of 30mg·min/l could be achieved with 1 mg/L residual after 30 minutes. Assuming the local lake water has a chlorine demand of 5 mg/l over 30 minutes, the chlorine dose should be 6 mg/l. Chlorinating a 20 l jerry can with household bleach (NaClO, 3% available chlorine) would therefore require 4 ml of bleach and a contact time of 30 minutes, assuming the bleach is mixed into the water. The dose can be linearly scaled with the available chlorine, e.g. a bleach with 1% available chlorine instead of 3% requires three times the dose. It should be noted that poorly produced bleach may lose stability due to improper packaging, exposure to heat and light during transportation or storage, or lack of pH stabilizers (pH should be 11–13) [45, 46]. A study evaluating the chlorine concentration in commercially available bleach in developing countries found that the advertised chlorine concentration deviated by an average 35% of the labelled concentration, highlighting the importance of quality control testing [45]. The chlorine demand and chlorine concentration could be measured by local WASH officials using a pool testing kit, for example, if available.

These chlorination recommendations can be applied to water infrastructure such as community showers, laundry stations or water recreation areas, thereby helping shift the community's water activities away from lakes and rivers and reducing contact with cercaria-infested water. Together with drug treatment, these facilities can reduce schistosomiasis incidence, as has been demonstrated by Kosinski *et al.* [6]. It is crucial that operators understand the concept of chlorine dosing to ensure that water is treated appropriately, and chlorine is adequately mixed into the water using manual or mechanical dosing. Chlorination is often preceded by filtration, such as filtration through a cloth or layer of sand. This removes organic particles, thereby reducing the water's chlorine demand and hence reducing the amount of chlorine that needs to be dosed. Chlorination is effective in inactivating other water-related diseases (e.g. *Vibrio cholerae* or *Escherichia coli*) found in the same waters as cercariae, as shown in Table 2. The similarity in CT values to those of other water-related diseases creates potential for providing safe water in a holistic approach–tackling several diseases with one treatment. While the provision of safe water and WASH facilities targeting schistosome cercariae could help reduce infection rates of other diseases, it may also have negative effects, for example if used as breeding sites for disease vectors such as mosquitoes. Water decisions should therefore always be made together with other local disease and WASH programs while keeping the community's needs in mind.

## Conclusions

This study has rigorously tested the effectiveness of chlorination against *S. mansoni* cercariae. Chlorination experiments indicate that *S. mansoni* cercariae are sensitive to chlorine, and higher chlorine CT values are required with higher water pH and lower temperature. A regression equation has been fit to the laboratory data, which can be used to predict CT values within

**Table 2. CT values for selected pathogens, adapted from Centers for Disease Control and Prevention [44]**

| Pathogen | CT (mg·min/l) | Inactivation | Temperature (˚C) | pH |
|---|---|---|---|---|
| *Escherichia coli* | 0.25 | 4-log | 23 | 7.0 |
| *Salmonella typhi* | 1 | 2-log | 20 | 7.0 |
| *Vibrio cholerae* (rugose strain) | 40 | 4-log | 20 | 7.0 |
| *Hepatitis A* | 0.41 | 4-log | 25 | 8.0 |
| *S. mansoni cercariae* (this study) | 30 | 2-log | 20 | 7.5 |

the tested pH and temperature range for 2-log inactivation of *S. mansoni* cercariae. Our results did not find a significant difference in chlorine sensitivities of laboratory-grown and field cercariae. Our most conservative results indicate that *S. mansoni* cercariae can be inactivated up to 2-log with a CT value of 26 mg·min/l at pH 7.5 and 20°C, though accounting for uncertainties leads to a CT value of 30 mg·min/l. A safety factor may be applicable if local conditions are unfavorable for effective chlorination (e.g. high turbidity, high cercarial concentration, pH > 7.5).

Improvements to water infrastructure are vital for attaining sustained control and elimination of schistosomiasis. Our findings can be used by households, water vendors and designers of community-scale infrastructure for water contact activities (e.g. laundry and bathing stations, showers) to treat infested water and provide safe, cercaria-free water facilities. We recommend that water treatment should always be accompanied by behavior change, to promote sustainable use of the infrastructure, as well as preventive chemotherapy, required to reduce disease prevalence. Strong collaboration between disease control programs and WASH stakeholders will be required to identify the most effective ways of integrating water infrastructure solutions [3, 47]. Together, these interventions can form an integral approach required to tackle this widespread neglected tropical disease.

## Acknowledgments

We thank Dr Teckla Angelo, Revocatus Alphonce, James Kubeja and Reuben Bugumba from the National Institute for Medical Research for their support in the laboratory, as we all undergraduate students Ottilie Shiyong Liu and Zixiang Chen for their help in setting up these experiments. We are also grateful to Prof Tony Walker at Kingston University and Dr Gabriel Rinaldi and Dr Matt Berriman of the Wellcome Sanger Institute for help with access to live schistosome material.

## Author Contributions

**Conceptualization:** Laura Braun, Fiona Allan.

**Data curation:** Laura Braun, Muluwork Maru.

**Formal analysis:** Laura Braun, Yasinta Daniel Sylivester, Michael R. Templeton.

**Funding acquisition:** Fiona Allan, Feleke Zewge, Aidan M. Emery, Safari Kinung'hi, Michael R. Templeton.

**Investigation:** Laura Braun, Yasinta Daniel Sylivester.

**Methodology:** Laura Braun, Yasinta Daniel Sylivester, Meseret Dessalegne Zerefa, Muluwork Maru, Michael R. Templeton.

**Resources:** Laura Braun, Fiona Allan, Feleke Zewge, Aidan M. Emery, Safari Kinung'hi, Michael R. Templeton.

**Supervision:** Fiona Allan, Safari Kinung'hi, Michael R. Templeton.

**Validation:** Laura Braun, Meseret Dessalegne Zerefa.

**Visualization:** Laura Braun.

**Writing – original draft:** Laura Braun, Michael R. Templeton.

**Writing – review & editing:** Laura Braun, Yasinta Daniel Sylivester, Fiona Allan, Feleke Zewge, Aidan M. Emery, Safari Kinung'hi, Michael R. Templeton.

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
