## [Decision Letter · Decision Letter 0]

17 Jul 2020

Dear Ms. Braun,

Thank you very much for submitting your manuscript "Chlorination of Schistosoma mansoni cercariae" for consideration at PLOS Neglected Tropical Diseases. As with all papers reviewed by the journal, your manuscript was reviewed by members of the editorial board and by several independent reviewers. The reviewers appreciated the attention to an important topic. Based on the reviews, we are likely to accept this manuscript for publication, providing that you modify the manuscript according to the review recommendations. 

Sincerely,

Xiao-Nong Zhou

Associate Editor

Simone Haeberlein

Deputy Editor

Reviewer's Responses to Questions

**Key Review Criteria Required for Acceptance?**

**Methods**

-Are the objectives of the study clearly articulated with a clear testable hypothesis stated?

-Is the study design appropriate to address the stated objectives?

-Is the population clearly described and appropriate for the hypothesis being tested?

-Is the sample size sufficient to ensure adequate power to address the hypothesis being tested?

-Were correct statistical analysis used to support conclusions?

-Are there concerns about ethical or regulatory requirements being met?

Reviewer #1: (No Response)

Reviewer #2: This is an excellently written paper with solid methods, results, and appropriately drawn conclusions.

Reviewer #3: (No Response)

**Results**

-Does the analysis presented match the analysis plan?

-Are the results clearly and completely presented?

-Are the figures (Tables, Images) of sufficient quality for clarity?

Reviewer #1: (No Response)

Reviewer #2: See comments.

Reviewer #3: (No Response)

**Conclusions**

-Are the conclusions supported by the data presented?

-Are the limitations of analysis clearly described?

-Do the authors discuss how these data can be helpful to advance our understanding of the topic under study?

-Is public health relevance addressed?

Reviewer #1: (No Response)

Reviewer #2: See comments.

Reviewer #3: (No Response)

**Editorial and Data Presentation Modifications?**

Reviewer #1: (No Response)

Reviewer #2: (No Response)

Reviewer #3: (No Response)

**Summary and General Comments**

Reviewer #1: The study entitled "Chlorination of Schistosoma mansoni cercariae " presents a comprehensive insight into treating infested water and provide safe, cercaria-free water facilities. The manuscript is well-written and the results are well documented. 

I have two major problems with presented arguments which the authors would need to clarify:

-It is considered that temperatures in Tanzania and elsewhere are sometimes below 20 degrees Celsius, the schistosomiasis cercariae will survive longer at low temperature and have better activity. It is of great significance to test the killing conditions of cercariae at low temperature. It is recommended to add the CT value of water chlorination at a lower temperature if possible.

-For drinking water standards, the final chlorine residue is an issue to consider. It is recommended that the study give the result of residual chlorine concentration after water treatment. Whether it is necessary to use chlorine removal before using the treated water, and what kind of dechlorination operation is appropriate.

Reviewer #2: Major comments

Section beginning on line 333. The comments on the health-relevance of 2 log10 reduction as a normative target for treatment are most welcome. This may be an arbitrary benchmark. I do not really understand this argument: 

“… water samples are unlikely to contain as high concentrations of cercariae as tested in our experiments (2 cercariae/ml). Therefore, a lower log inactivation may be sufficient.”

The authors rightly state just above that “water should be completely free of cercariae” (lines 338 and 339). The relevance of the 2 log reduction target, in terms of exposure, is unclear, since one cercaria = one schistosome and volume is relevant here because exposure is to volumes of water. Let’s take 1000 L of water. We add in 1000 cercariae, so 1 cercaria per L (far less concentrated than the experimental conditions of 2 cercariae per ml). We dose this to achieve 2 log reduction, according to the recommendations here. That leaves 10 cercaria in the volume. Would that produce any exposure risk? Would the risk be more tolerable, and by what standard? I’m grappling with what conclusions we can draw about the likelihood of the recommended treatment to reduce risk in endemic settings. Can the authors present a mechanistic argument for why a 2 log reduction is sufficient or even, as they suggest, conservative? Certainly fewer cercariae would be better than more, but how do we know this would be “good enough”, and how is “good enough” defined? 

Minor comments

Line 27. Add log base. From the text, it’s clear this is log10, but should be specified. 

Line 349 and beyond. It should be acknowledged that most chlorine available in endemic settings would be of uncertain concentration to most users. Therefore the problem of measuring chlorine remains. CT values are valuable, but doses can only be shown to meet the target if we know what concentration of chlorine we have to begin with. Household bleach comes in a variety of concentrations and cannot be assumed in most settings. Empirical calibration of the dose, measured by a pool tester (at least), is probably the most feasible option in most settings.

Reviewer #3: This study investigates the effectiveness of chlorination against S. mansoni cercariae. Chlorination. The experiments indicate that S. mansoni cercariae can be inactivated up to 2-log with a CT value of 26 mg·min/l at pH 7.5 and 20°C, though adding in a safety factor to account for uncertainties would lead to a recommended CT value of 30 mg·min/l. It means that S. mansoni cercariae are sensitive to chlorine, and higher chlorine CT values are required with higher pH and lower temperature. The paper reads well, while important preliminary findings. One comment–in Discussion, the results presented here can be used as the basis for water treatment guidelines for S. mansoni infested water. How would the author suggest this treatment would be carried out? Because the results did not find a significant difference in chlorine sensitivities of laboratory-grown and field cercariae.

PLOS authors have the option to publish the peer review history of their article (what does this mean?). If published, this will include your full peer review and any attached files.

Reviewer #1: No

Reviewer #2: No

Reviewer #3: Yes: Zhiqiang Qin
---

## [Editor Report · Decision Letter 1]

1 Aug 2020

Dear Ms. Braun,

We are pleased to inform you that your manuscript 'Chlorination of Schistosoma mansoni cercariae' has been provisionally accepted for publication in PLOS Neglected Tropical Diseases.

Best regards,

Xiao-Nong Zhou

Associate Editor

Simone Haeberlein

Deputy Editor

---

## [Editor Report · Acceptance letter]

14 Aug 2020

Dear Ms. Braun,

We are delighted to inform you that your manuscript, "Chlorination of *Schistosoma mansoni* cercariae," has been formally accepted for publication in PLOS Neglected Tropical Diseases.

Best regards,

Shaden Kamhawi

co-Editor-in-Chief

Paul Brindley

co-Editor-in-Chief
